# The Effects of In-Plane Spatial Resolution on CT-Based Radiomic Features’ Stability with and without ComBat Harmonization

**DOI:** 10.3390/cancers13081848

**Published:** 2021-04-13

**Authors:** Abdalla Ibrahim, Turkey Refaee, Sergey Primakov, Bruno Barufaldi, Raymond J. Acciavatti, Renée W. Y. Granzier, Roland Hustinx, Felix M. Mottaghy, Henry C. Woodruff, Joachim E. Wildberger, Philippe Lambin, Andrew D. A. Maidment

**Affiliations:** 1The D-Lab, Department of Precision Medicine, GROW—School for Oncology, Maastricht University, 6200 Maastricht, The Netherlands; t.refaee@maastrichtuniversity.nl (T.R.); S.primakov@maastrichtuniversity.nl (S.P.); h.woodruff@maastrichtuniversity.nl (H.C.W.); philippe.lambin@maastrichtuniversity.nl (P.L.); 2Department of Radiology and Nuclear Medicine, Maastricht University Medical Centre+, 6200 Maastricht, The Netherlands; felix.mottaghy@mumc.nl (F.M.M.); j.wildberger@mumc.nl (J.E.W.); 3Division of Nuclear Medicine and Oncological Imaging, Department of Medical Physics, University Hospital of Liège and GIGA CRC-In Vivo Imaging, University of Liège, 4000 Liege, Belgium; rhustinx@ulg.ac.be; 4Department of Nuclear Medicine and Comprehensive Diagnostic Center Aachen (CDCA), University Hospital RWTH Aachen University, 52074 Aachen, Germany; 5Department of Diagnostic Radiology, Faculty of Applied Medical Sciences, Jazan University, Jazan 45142, Saudi Arabia; 6Department of Radiology, Perelman School of Medicine, University of Pennsylvania, Philadelphia, PA 19104, USA; Bruno.Barufaldi@pennmedicine.upenn.edu (B.B.); racci@pennmedicine.upenn.edu (R.J.A.); Andrew.Maidment@pennmedicine.upenn.edu (A.D.A.M.); 7Department of Surgery, GROW—School for Oncology, Maastricht University Medical Centre+, 6200 Maastricht, The Netherlands; r.granzier@maastrichtuniversity.nl

**Keywords:** image processing, harmonization, reproducibility, radiomics biomarkers

## Abstract

**Simple Summary:**

Handcrafted radiomic features (HRFs) are quantitative features extracted from medical images, and they are mined for associations with different clinical endpoints. While many studies reported on the potential of HRFs to unravel clinical endpoints, the sensitivity of HRFs to variations in scanning parameters is affecting the inclusion of radiomic signatures in clinical decision-making. In this study, we investigated the effects of variations in the in-plane resolution of scans, while all other scanning parameters were fixed. Moreover, we investigated the effects of ten different image resampling methods and ComBat harmonization on the reproducibility of HRFs. Our results show that the majority of HRFs are significantly and variably affected by the differences in in-plane resolution. The majority of image resampling methods resulted in a higher number of reproducible HRFs compared to ComBat harmonization. Our developed framework guides identifying the reproducible and harmonizable HRFs in different scenarios.

**Abstract:**

While handcrafted radiomic features (HRFs) have shown promise in the field of personalized medicine, many hurdles hinder its incorporation into clinical practice, including but not limited to their sensitivity to differences in acquisition and reconstruction parameters. In this study, we evaluated the effects of differences in in-plane spatial resolution (IPR) on HRFs, using a phantom dataset (n = 14) acquired on two scanner models. Furthermore, we assessed the effects of interpolation methods (IMs), the choice of a new unified in-plane resolution (NUIR), and ComBat harmonization on the reproducibility of HRFs. The reproducibility of HRFs was significantly affected by variations in IPR, with pairwise concordant HRFs, as measured by the concordance correlation coefficient (CCC), ranging from 42% to 95%. The number of concordant HRFs (CCC > 0.9) after resampling varied depending on (i) the scanner model, (ii) the IM, and (iii) the NUIR. The number of concordant HRFs after ComBat harmonization depended on the variations between the batches harmonized. The majority of IMs resulted in a higher number of concordant HRFs compared to ComBat harmonization, and the combination of IMs and ComBat harmonization did not yield a significant benefit. Our developed framework can be used to assess the reproducibility and harmonizability of RFs.

## 1. Introduction

In recent years, quantitative medical imaging research using handcrafted radiomic features (HRFs) has been growing exponentially [1,2]. Radiomics refers to the high throughput extraction of quantitative imaging features that are expected to correlate with clinical and biological characteristics of patients [3,4]. For decades, it has been hypothesized that image texture analysis could potentially extract more information from a region of interest (ROI) than that solely perceived by the human eye [5,6]. Yet, the term radiomics has only been introduced recently [7,8]. HRFs are generally grouped into shape, intensity, and textural features. To date, many studies have reported on the potential of radiomics to predict various clinical endpoints [9,10]. However, major challenges, including the reproducibility of the HRFs across different acquisition and reconstruction parameters, have hindered the incorporation of radiomics in clinical decision support systems [11,12]. 

The essence of radiomics is that certain HRFs help decode biologic information [8], allowing these features to be treated as biomarkers. The mainstay of a biomarker is the ability to quantify it in a reproducible manner [13]. HRFs are mathematical equations applied to numeric arrays of intensity values that form the medical image. Therefore, it is intuitive that changes in the values in the array (due to differences in scan acquisition and reconstruction parameters), by the transitive property, lead to (potentially significant) quantitative changes in the HRFs. It is well established that changes in scan acquisition and reconstruction parameters affect the values in the array representing the medical image [14]. Therefore, it is a common clinical practice to scan a phantom to calibrate the CT scanner on a routine basis. Hence, similar practices are needed before radiomics studies are conducted, when the scans under analysis were acquired using heterogeneous acquisition and reconstruction parameters [15]. Many studies have already reported on the sensitivity of HRFs to different factors, including (i) temporal variability, or test–retest [16,17], in which two scans of a patient (or a phantom) are taken after a time interval using the exact scanning parameters; and (ii) scanning parameters variability [11,18,19], in which an object (usually a phantom) is scanned multiple times using different scanning parameters. Variations in the majority of scanner/scanning parameter combinations were reported to impact the reproducibility of HRFs significantly [18,19,20].

One scan reconstruction parameter expected to have an effect on the reproducibility of HRFs is the in-plane spatial resolution (IPR), which is dictated in part by the pixel dimensions, while the through-plane spatial resolution is determined by the slice thickness and slice spacing. Resampling all the scans in a dataset to a new unified in-plane spatial resolution (NUIR) before feature extraction has been employed as a method to reduce the variation in radiomic feature values [21,22]. The NUIR is usually decided based on the most frequent IPR in the dataset, and different interpolation methods (IMs) can be used for this purpose. Interpolation is a model-based method to recover continuous data from discrete data within a known range of data spacings (i.e., pixel size in images) [23]. The degree to which data recovery is possible is highly sensitive to the interpolation method and the underlying data structure. In the case of medical imaging analysis, interpolation is employed either to convert the spatial sampling rate (measured in pixels or voxel count per unit of length per dimension) to another or to distort the image in the case of image registration [24]. Since the vast majority of HRFs are derived from pixel/voxel values and their distributions, interpolation to a common pixel spacing could potentially reduce variance introduced to these HRFs arising from differences in IPR.

As a rule, one must distinguish between interpolation methods that increase or reduce the image resolution. Interpolation from smaller pixels to larger pixels (i.e., reducing spatial resolution) usually involves some form of averaging, with the possible exception of modern deep learning-based methods.

Generally, while data acquired with small pixels will contain more noise, the process of averaging to large pixels will ameliorate the noise properties. As such, the process is less sensitive to the interpolation method/model. On the other hand, interpolation from larger pixels to smaller pixels (i.e., increasing spatial resolution) is fraught with challenges, as the interpolated data can be highly sensitive to the interpolation model due to the need to create de novo pixel values. Larger pixels average the signal over a larger area than smaller ones, leading to the loss of variations in the original scene that occur over spatial frequencies smaller than the Nyquist limit and cannot be recovered exactly.

Certain methods, such as nearest neighbor interpolation (also called pixel replication), while fast, are less accurate than other methods such as sinc interpolation or deep-learning methods (which are trained with representative data). However, all such interpolation methods are sensitive to biases arising from the image [25]. The application of these methods to medical imaging has been evaluated qualitatively [26]. Yet, the effects of these methods on the reproducibility of HRFs is not well understood. Unlike humans, whose exposure to a vast assortment of scanners, patients, and acquisition conditions (including IPR) leads to a tolerance for such changes, IPR is likely to have more profound effects on HRFs.

A harmonization method that has become increasingly common in the field of radiomics is ComBat. ComBat was originally developed for the harmonization of gene expression arrays [27]. Several studies have investigated the potential of ComBat in radiomics analysis and recommended its use [28,29]. We hypothesize that ComBat, the chosen IM, and the selected NUIR will affect the reproducibility of HRFs differently. In this study, the reproducibility of HRFs was assessed across different IPRs, while keeping all other parameters fixed, using a public dataset of CT scans of a phantom. A thorough investigation of the applicability of 10 different IMs was performed in an effort to identify suitable IMs for the purpose of increasing the number of reproducible HRFs in a heterogeneous dataset. In particular, we investigated whether data with discordant pixel sizes need to be interpolated to a common pixel size to perform radiomics analysis, and we also investigated how the choice of IM and NUIR, as well as ComBat harmonization, affects the reproducibility of HRFs. Furthermore, we developed a generalizable workflow that assesses the impact of different harmonization techniques (Figure 1) on the reproducibility of RFs. Ultimately, the goal of our work is to guide robust radiomics analysis to ease its incorporation in clinical decision-making.

## 2. Materials and Methods

### 2.1. Phantom Data

The publicly available Credence Cartridge Radiomics (CCR) phantom data [30] found in The Cancer Imaging Archive (TCIA.org) [31] was used. The CCR phantom is composed of 10 different layers that correspond to different texture patterns spanning a range of almost −900 to +700 HU (Appendix A). The publicly available dataset includes 251 scans of the phantom acquired using six scanner models manufactured by three different manufacturers. The scans were acquired using various acquisition and reconstruction parameters to assess the reproducibility of HRFs. For the purpose of this study, 14 scans acquired using 2 different scanner models (Discovery STE & LightSpeed Pro 32) of the same manufacturer (GE), which were all acquired at a single slice thickness (1.25 mm), tube voltage (120 kV), tube current (250 mA), and convolution kernel (standard), but varying IPR (Table 1) were used. The reasoning behind this selection is multifold: (i) the effects of the variations are expected to be dependent on the heterogeneity in acquisition; (ii) the number and complexity of the different combinations available are too huge to be described, analyzed, and presented in a single experiment; (iii) the data under analysis were acquired using the same scanner models, and the same acquisition and reconstruction parameters except for the in-plane resolution, which allows the assessment of the effect of variations in this single parameter.

### 2.2. Interpolation and Image Resampling

The effects of the IMs included in the popular open-source radiomics toolbox PyRadiomics [32] were assessed in this study. The methods are based on the python library Simple-ITK [33], and they include (i) nearest neighbor (NN), (ii) linear, (iii) basis spline (B-spline), (iv) Gaussian, (v) Gaussian using labelling (mask) information (LabelGaussian), and windowed sinc interpolations using the following window types: (vi) Hamming (HammingWindowedSinc or HWS), (vii) Cosine (CosineWindowedSinc or CWS), (viii) Welch (WelchWindowedSinc or WWS), (ix) Lanczos (LanczosWindowedSinc or LWS), and (x) Blackman (BlackmanWindowedSinc or BWS).

The simplest of these IMs, and the ones with the lowest computational costs, are (i) the NN interpolation, which functions by assigning any new voxel the same value as its closest neighbor in the original image; and (ii) linear interpolation, in which the values of new pixels are interpolated linearly between the two original values [26]. B-spline interpolation is more complex than NN or linear; the calculations span four pixels [34]. While the method performs well in terms of radiologic evaluation in which the aim is to convince human observers, it is known to unnecessarily over-smooth the image [26]. The windowed sinc functions are complex convolution-based interpolations that are based on multiplying the sinc function by a limited spatial support window to reduce unwanted effects on the resampled image [35], which is followed by filtering of the frequencies to avoid the injection of spurious frequency components. Windowed sinc functions are generally considered superior to other interpolation methods, as little superfluous noise is injected into the interpolated images.

### 2.3. HRFs Extraction

Each scan contained 10 independent regions of interest (ROIs) (one for each layer of the phantom) that occupy the same physical area of the phantom on each scan. For each ROI, HRFs were calculated using the open source software Pyradiomics V 2.1.2. HRFs were extracted multiple times to perform different experiments. First, to assess the effect of differences in in-plane resolution and ComBat harmonization on HRFs, no changes to the original in-plane resolution were made. Second, to assess the effect of different IMs and NUIRs and the combination of interpolation and ComBat, HRFs were extracted from the scans using all IMs and all available NUIRs in the dataset (Table 1).

For each set of scans (7 scans, with 10 ROIs per scan) from each scanner model (n = 2), HRFs were extracted 71 times. The HRFs were extracted one time from the original scans and 70 times with unique combinations of IM and NUIR. In each run, a total of 91 original RFs were extracted. In Pyradiomics, shape features are calculated on the original input image, and they are not affected by the in-application resampling. Therefore, those HRFs were excluded. 

To reduce noise and computational requirements, images were pre-processed by binning voxel grayscale values into bins with a fixed width of 25 Hounsfield units (HUs) for extracting HRFs from unfiltered images. No other image pre-processing steps were performed. The extracted HRFs included HU intensity features, and texture features describing the spatial distribution of voxel intensities using 5 texture matrices (gray-level co-occurrence (GLCM), gray-level run-length (GLRLM), gray-level size-zone (GLSZM), gray-level dependence (GLDM), and neighborhood gray-tone difference (NGTDM) matrices). A more detailed description of the Pyradiomics HRFs can be found online (https://pyradiomics.readthedocs.io/en/latest/features.html (accessed on 7 January 2021)).

### 2.4. ComBat Harmonization

ComBat is an empirical Bayes based method used to estimate the effects of different batches on HRFs; in this scenario, variations in scan acquisition and reconstruction parameters were considered [27]. The ComBat method assumes that a feature value can be approximated by the equation:(1)$$Y_{ij}=\alpha +\beta X_{ij}+\gamma_{i}+\delta_{i}\varepsilon_{ij}$$
where α is the average value for feature $Y_{ij}$ for ROI *j* on scanner *i*; *X* is a design matrix of the biologic covariates known to affect the HRFs; β is the vector of regression coefficients corresponding to each biologic covariate; $\gamma_{i}$ is the additive effect of scanner *i* on HRFs, $\delta_{i}$ is the multiplicative scanner effect, and *ε_ij_* is an error term, which is presupposed to be normally distributed with zero mean. Based on the values estimated, ComBat performs feature transformation in the form of:(2)$$Y_{ij}^{ComBat}=\frac{\left(Y_{ij}-\hat{\alpha }-\hat{\beta }X_{ij}-\gamma_{i}^{*}\right)}{\delta_{i}^{*}}+\hat{a}+\hat{\beta }X_{ij}$$
where $\hat{\alpha }$ and $\hat{\beta }$ are estimators of parameters *α* and *β*, respectively. $\gamma_{i}^{*}$ and $\delta_{i}^{*}$ are the empirical Bayes estimates of $\gamma_{i}$ and $\delta_{i}$, respectively [28].

### 2.5. Statistical Analysis

To assess the agreement of a given HRF for the same ROI scanned using different settings and scanners, the concordance correlation coefficient (CCC) was calculated using the epiR package (Version 0.9–99) [36] and R language (Version 3.5.1) [37] with R studio (Version 1.1.456) [38]. The CCC is used to evaluate the agreement between paired readings [38], and it provides the measure of concordance as a value between 1 and −1, where 0 represents no concordance and 1 or −1 represent a perfect direct positive or inverse concordance, respectively. The CCC metric further has the advantages of (i) robustness in small sample sizes and (ii) taking the rank and value of the feature into consideration [39]. The cut-off of (CCC > 0.9) was used to select reproducible HRFs, as the literature suggests that values < 0.9 indicate poor concordance [40].

Four different approaches for assessing concordances of HRFs were used (Figure 2): (i) HRFs extracted from the original scans; (ii) HRFs extracted from the original scans and harmonized using ComBat; (iii) HRFs extracted from resampled scans; and (iv) HRFs extracted from resampled scans harmonized using ComBat. For (i), the CCC was calculated for all HRFs of all ROIs across 7 different scans from each scanner. In each run, the CCC was calculated between a different pair of scans. For (ii), HRFs with nearly zero variance (i.e., HRFs which have the same value in 95% or more of the data points) had to be removed before applying ComBat. Parametric prior estimations were used, and no reference batch was assigned for ComBat application. CCC was calculated after harmonizing the remaining HRFs using ComBat. In each run, ComBat was applied on two batches (scans). For (iii), the CCC was calculated for the HRFs following feature extraction with each of the IMs. The effects of the NUIR were assessed by calculating the CCC for the HRFs after resampling all the scans to one of the available in-plane resolutions. For (iv), ComBat was applied after the same process in (iii), and then, the CCC was calculated. To gauge an overall image of the reproducibility of HRFs across all pairs as well as the impact of IMs, NUIRs, and ComBat, the number (percentage) of HRFs that were reproducible by taking the intersection of HRFs that were reproducible in each pairwise comparison of a certain scenario were compared (21 pairs in each scenario as shown in Table 2, Table 3, Table 4 and Table 5).

Furthermore, we assessed the correlation between the HRFs that were concordant across all pairwise comparisons on each scanner model, using Spearman correlation [41]. HRFs were considered highly correlated if the Spearman’s correlation coefficient had a value > 0.90.

## 3. Results

### 3.1. Approach (i): Effects of IPR on the Reproducibility of HRFs

The number of HRFs insensitive to the variations in IPR depended on the scanner model (Table 2 and Appendix A). In pairwise comparisons, the number of concordant HRFs was lower when the difference in IPR between the scan pairs was greater. The lowest concordance was observed between the scan with the highest resolution and the scan with the lowest resolution.

Out of the 91 extracted HRFs, between 39 (42.9%) and 86 (94.5%) HRFs were concordant, varying pairwise and scanner wise. Some HRFs were robust to variations in IPR in one scanner model and not in the other.

On the Discovery STE model (GE), the number of concordant HRFs ranged between 39 (42.9%) and 86 (94.5%), with a median of 70 (39.6%) HRFs (Table 2). A total of 36 (39.6%) HRFs were reproducible regardless of the IPR selected when all other scanning parameters were fixed (Appendix A). Of these 36 HRFs, nine remained after removing highly correlated HRFs (Appendix A), and none was highly correlated with volume. Overall, the Lightspeed Pro 32 model showed lower concordance than the Discovery STE model. The number of pairwise concordant HRFs on the Lightspeed Pro 32 model ranged between 39 (42.8%) and 82 (90.1%), with a median of 60 (65.9%) (Appendix A). A total of 27 (29.7%) HRFs were reproducible across all pairs (Appendix A). Of these 27 HRFs, nine remained after removing highly correlated HRFs (Appendix A), and none was highly correlated with volume. Twenty-six (28.6%) HRFs were reproducible on both scanner models regardless of the IPR.

### 3.2. Approach (ii): ComBat Harmonization of HRFs Extracted from Original Scans

ComBat harmonization increased the number of concordant HRFs compared to before harmonization. On the Discovery model, the increment in the number (percentage) of HRFs ranged between 0 (0%) and 13 (14.3%), with a median of 6 (6.6%) of the total depending on the batches being harmonized (Table 3). 46 (50.5%) HRFs were found to be reproducible across all pairwise comparisons following ComBat harmonization, 35 of which were found to be highly correlated. The number of concordant HRFs decreased with the increment in IPR variation. Hence, the increment in the number of concordant HRFs was larger when the batches being harmonized had a larger difference in IPR.

The performance of ComBat had a similar pattern on both the Discovery STE and the Lightspeed Pro 32 models. The increment in the number (percentage) of concordant HRFs extracted from the scans acquired with the Lightspeed Pro 32 model following ComBat harmonization ranged between 1 (1.1%) and 14 (15.4%) HRFs with a median increment of 7 (7.7%) HRFs compared to before harmonization, depending on the batches being harmonized (Appendix A). Forty-one (45.1%) HRFs were reproducible across all pairs following ComBat harmonization, 29 of which were found to be highly correlated.

### 3.3. Approach (iii): The Effects of Different IMs and NUIR on HRFs

Different interpolation methods showed different effects on the reproducibility of HRFs. These effects further depended on the selected NUIR and the scanner model (Figure 3 and Appendix A). For the majority of combinations of scanner models, IMs and NUIRs, some HRFs were only concordant when extracted from the original scans; some HRFs became concordant only after resampling, while some lost their concordance following resampling (Appendix A). CSW resampling to the highest and lowest resolutions are used below as detailed examples on both scanner models.

On the Discovery STE model, the use of windowed sinc IMs resulted in an overall increment in the number of reproducible HRFs, regardless of the NUIR selected. The range of HRFs that had an improved concordance across all pairs when using windowing sinc was between 14 (15.4%) and 20 (22%) HRFs, depending on the NUIR. When scans were resampled to the highest resolution using CWS, the increment in the number of concordant HRFs ranged between −2 (−2.2%) and 36 (39.6%), with a median of 12 (13.2%) HRFs. Moreover, 47 (51.6%) HRFs were concordant across all pairs. When scans were resampled to the lowest resolution using CWS, the increment in the number of concordant HRFs ranged between 4 (4.4%) and 35 (38.5%), with a median of 16 (17.6%) HRFs. 54 (59.3%) HRFs were concordant across all pairs. Table 4 shows the pairwise number (percentage) of reproducible HRFs following resampling to the median IPR value with CWS IM on the Discovery model for comparison with Table 5.

HWS performed the best when the images were resampled to a NUIR equal to or lower than the median (0.49 × 0.49 mm^2^), while CWS, WWS, and LWS methods performed better on NUIR values higher than the median. BSpline IM resulted in a minor to significant increment in the number of reproducible HRFs, with a higher number of concordant features when higher NUIRs where chosen. Gaussian and Label-Gaussian IMs consistently resulted in lower numbers of concordant HRFs. The number of HRFs losing concordance across all pairs when using a Gaussian IM ranged between −29 (−31.9%) and −30 (−33%) HRFs, while the range for LabelGaussian was between −11 (−12.1%) and −19 (−20.9%) HRFs, depending on the NUIR. The rest of IMs (NN and Linear) resulted in an overall decrease in the number of concordant HRFs when an NUIR below the median resolution was selected, and there was a minor-significant improvement with NUIRs higher than the median resolution (Appendix A).

On the Lightspeed Pro 32 model, windowed sinc IMs (except for BWS) showed a consistent increment in the number of reproducible HRFs, which was varying depending on the NUIR. When scans were resampled to the highest resolution using CWS, the increment in the number of concordant HRFs ranged between −9 (−9.9%) and 36 (39.6%), with a median of 8 (8.8%) HRFs. Thirty (33%) HRFs were concordant across all pairs. When scans were resampled to the lowest resolution using CWS, the increment in the number of concordant HRFs ranged between −3 (−3.3%) and 31 (34.1%), with a median of 16 (17.6%) HRFs. 38 (41.8 %) HRFs were concordant across all pairs. Appendix A shows the pairwise number (percentage) of concordant HRFs following resampling to the median IPR value with CWS IM on the LightSpeed Pro 32 model, for comparison with Appendix A. The application of other IMs (BWS, NN, Linear, Gaussian, and Label-Gaussian) with an NUIR other than the two lowest resolutions available resulted in an overall decrease in the number of concordant HRFs. However, when the lowest resolution was selected as NUIR, BSpline IM outperformed all other methods when the number of concordant HRFs across all pairs was considered (Appendix A).

### 3.4. Approach (iv): The Combination of IMs and ComBat Harmonization

Approach (iii) resulted in a higher number of concordant HRFs in the majority of pairwise scenarios compared to approach (ii) for the majority of IMs that performed solely well (for example, Table 3 vs. Table 4). The application of ComBat harmonization on HRFs extracted from resampled scans varied per scanner model, IMs, NUIRs, and batches. However, when the number of concordant HRFs across all pairs is considered, ComBat increased the number of concordant HRFs in almost all of the investigated scenarios (Figure 4 and Appendix A).

On the Discovery model, the increment in the number (percentage) of concordant HRFs extracted from scans resampled to the highest resolution after ComBat harmonization ranged between 0 (0%) and 10 (11%), with a median increment of 0 (0%) of the total number of HRFs compared to before harmonization. Fifty-four (59.3%) HRFs were concordant across all pairs. When ComBat was applied on HRFs extracted from scans resampled to the lowest resolution, the increment in the number (percentage) of HRFs ranged between −1 (−1.1%) and 10 (11%) HRFs, with a median of 0 (0%), depending on the batches being harmonized. Sixty-one (67%) were found to be stable across all pairs. Table 5 shows the number of pairwise concordant HRFs following the application of ComBat on scans acquired on the Discovery STE model and resampled to the median IPR value using CWS IM.

On the LightSpeed Pro 32 model, the increment in the number (percentage) of concordant HRFs after ComBat harmonization on HRFs extracted from scans resampled to the highest resolution (lowest concordance) ranged between −1 (−1.1%) and 13 (14.3%) HRFs, with a median of 3 (3.3%) of the total number of HRFs compared to before harmonization. Forty-two (46.2%) HRFs were concordant across all pairs. When ComBat was applied on HRFs extracted from scans resampled to the lowest resolution (highest concordance), the increment in the number (percentage) of HRFs ranged between 0 (0%) and 10 (11%) HRFs, with a median increment of 1 (1.1%) feature. Fifty-one (56%) HRFs were concordant across all pairs. Appendix A shows the pairwise CCC following the application of ComBat on scans acquired with the LightSpeed Pro 32 model and resampled to the median IPR value using CWS IM.

## 4. Discussion

In this study, the effects of variations in scans’ IPR on the reproducibility of HRFs, the proper methodology of identifying HRFs that are reproducible across different IPRs, and how to properly adjust for these differences before performing radiomics analysis using image interpolation and/or ComBat harmonization were thoroughly investigated. Uniquely, this study evaluates the effects of all the different IMs and the choice of NUIRs on the reproducibility of HRFs. Previous studies usually investigated a single IM with a single NUIR [21,22].

While two batches of scans acquired with the same imaging parameters on two scanner models of the same vendor were used for analysis, the effects of IPR, ComBat, IMs, and NUIR on the reproducibility of HRFs varied on each of the scanner models. The CCC was calculated pairwise to assess the reproducibility of HRFs when different sets of data were used as batches. Calculating the pairwise CCC between HRF values extracted before resampling the images revealed that the reproducibility of HRFs in our data depended on several factors including, but not limited to, the definition of the HRF, the degree of variation in IPR, and the scanner (hardware) make/model. Addressing the effects of these factors is crucial for performing robust radiomics analysis.

Without performing image preprocessing, the number of reproducible HRFs varied according to the batches being assessed. The aim of this study was to show that different investigated scenarios showed different numbers of reproducible HRFs. Therefore, although 36 HRFs for the Discovery STE scanner (27 HRFs for LightSpeed Pro 32 scanner) were always included in the set of concordant HRFs, it is difficult to conclude that these HRFs are insensitive to spatial resolution on all other scanner models based on our experiments. Yet, our framework guides the methodology of identifying reproducible HRFs according to the data under analysis. As we have shown, the number and type of HRFs is at least sensitive to the scanner model by the same manufacturer. Moreover, we anticipate based on their definition that certain HRFs (such as histogram-based features) are less sensitive, while others (e.g., texture features) are more sensitive to variations in scanning parameters and/or imaging vendors. Generally, scans with more similar original IPRs, and those of integer multiples of IPR showed higher numbers of concordant HRFs before and after resampling. This can be explained by the mechanisms by which a scan is acquired. When all other scanning parameters are fixed, the variations in IPR will result in variations in the number of pixels in 2D, while the other dimensions are preserved. Therefore, when all other parameters are fixed, the closer the IPR values are, the closer the values of the extracted HRFs.

For the IMs, the number of HRFs that had better/worse concordance after resampling was dependent on the NUIR chosen and scanner model. The window sinc interpolation family performed consistently better on both scanners and NUIRs investigated. In the field of radiology, both NN and linear are known to result in imprecisions [26,35]. A study into the reproducibility of HRFs investigated the performance of B-spline, linear, and NN using a single image slice thickness, and it concluded that NN is not a favorable method for the reproducibility of HRFs [42]. Our results support these previous reports by showing that NN and linear IMs are not the best candidates for improving the reproducibility of HRFs among scans acquired with different IPRs, and their use led to lower numbers of concordant HRFs in many of the investigated scenarios.

With regard to the selection of NUIR, a common trend of an inverse relationship between the NUIR and the number of concordant HRFs following resampling was observed. This trend was observed in both scanner models investigated. However, the percentage difference between the concordant HRFs is not significant at the lower end of the NUIR spectrum (Figure 3, Figure 4, Appendix A). As the best NUIR is expected to be task dependent (for e.g., classification of a lesion, predicting response to therapy or overall survival, etc.), outcome-based analysis is needed to determine the best NUIR. Yet, as a general rule, the smaller the NUIR, the better the concordance. In addition, while the number of non-highly correlated HRFs was found to be low on both scanner models (nine and 11 HRFs before and after ComBat harmonization, respectively), the exclusion of highly correlated HRFs should be performed based on the effects of the removal of these HRFs on the model performance.

A previous study investigated the effects on HRFs of voxel size resampling using linear interpolation. The authors resampled the scans of a phantom to a single voxel size, which was larger than the largest voxel size in the original scans, and it reported that around 20% of the HRFs (n = 213) became concordant after resampling [22]. Another study also investigated the effects of voxel size on HRFs of lung cancer patients [21]. The authors resampled all the scans to a single common voxel size using linear interpolation, and they reported that resampling does not eliminate all the variations in feature values even when the only variation in scan acquisition and reconstruction parameters was the voxel size, but it is favorable to no resampling. Another group investigated the effects of variation in several acquisition and reconstruction parameters on a 13-layer phantom using a different approach, and they reported that resampling the scans to isotropic voxels increased the percentage of concordant HRFs from 59.5% to 89.3% [43]. In this study, we found a similar conclusion: the number of previously non-concordant HRFs that became concordant following resampling to the lowest resolution ranged between 1.1% and 22% depending on the IM, and not all HRFs benefit from image resampling.

In contrast to previous studies, we investigated more IMs and harmonization techniques, and we propose a guideline on how to carefully approach HRFs reproducibility studies. Furthermore, we found that linear interpolation is not a good candidate for the purpose of improving the reproducibility of HRFs when compared to other available IMs, and the performance of an IM is dependent on the original IPR values and the chosen NUIR as well as the imaging vendor. 

When pairwise comparisons were considered, the performance of ComBat harmonization was found to be inferior to that of well-performing IMs, regardless of the NUIR. Moreover, the combination of ComBat and the well-performing IMs did not yield significantly better results compared to solely using the IM. Furthermore, the performance of ComBat varied depending on the batches used. Nevertheless, when the number of concordant HRFs across all pairs was considered, ComBat harmonization was of added value in almost all scenarios. Therefore, ComBat application on HRFs should follow a reproducibility study (phantom or tissue studies) to assess the impact of ComBat on the reproducibility of HRFs in those settings and use only the harmonizable HRFs for further radiomics analyses [15], as described in the workflow (Figure 1). The application of ComBat without assessing HRFs’ reproducibility as described may result in the inclusion of a high percentage of unreproducible HRFs or even the loss of some of the HRFs that were originally reproducible, rendering the analysis of these HRFs meaningless. This finding regarding ComBat harmonization is not in line with previous reports, which reported that ComBat successfully removes the batch effects for all HRFs [28,44]. This could be attributed to the differences in the radiomics software and/or the evaluation metrics used. In contrast to previous studies, and as the aim of harmonization is to improve reproducibility but not necessarily the performance of generated radiomic models, we opted for the CCC. The CCC provides an accurate description of the reproducibility of HRFs, which is not reflected in neither the distribution of HRFs nor the performance of radiomics models [45]. If radiomic models are to be used clinically, it is expected to be applied to one patient per time. Therefore, the importance has been given in this study to the individual feature values and not their distributions. HRFs with different values and order rank can share similar distributions, in which case the feature cannot be considered reproducible. In addition, different modeling techniques may yield significantly different results on the same dataset. Hence, the difference in the performance of a radiomic signature before and after harmonization does not necessarily inform about the performance of the harmonization method. Our proposed framework addresses this issue and guides the selection of reproducible and harmonizable HRFs before developing a radiomic signature, which helps the translation and generalization of results, and ultimately the inclusion of radiomic signatures in clinical practice.

Of note, not all HRFs benefit from resampling all scans to a NUIR or using ComBat harmonization. Some HRFs lost their concordance following resampling, depending on the IM employed and the chosen NUIR. The combination of IMs and NUIRs affected the HRFs differently on different scanner models. Some HRFs were not found to be concordant on one of the scanner models before or after resampling to any of the available NUIRs using any of the IMs, but they were found to be concordant on the other scanner model. Other HRFs were found to be concordant across different scanner models and IPRs. These findings indicate the need for performing reproducibility studies depending on the data under study, and the fact that at this level, we are unable to provide a list of HRFs that can be used regardless of the acquisition and reconstruction parameters and scanner models used. However, it lays down the basis for identifying reproducible HRFs before performing data analysis. In real-life scenarios, the variations between the imaging parameters in retrospective cohorts (especially multicentric) are usually not only limited to the IPR. Aside from the scanner/scanning parameters combination variations, some of the effects will be attributed to patient populations. Furthermore, while phantom studies reflect on the reproducibility of HRFs extracted from anthropomorphic phantoms, HRFs extracted from human tissue are expected to have a wider range of variations due to the inclusion of biologic factors. This knowledge, combined with our findings, necessitate the critical investigation of the reproducibility of HRFs across the different scanning parameters/scanners before performing any statistical analysis, and future investigations into the effects of differences in acquisition and reconstruction parameters on the reproducibility of HRFs extracted from human tissues, if feasible. Directly performing radiomics analysis on data acquired heterogeneously leads to spurious results and lacks meaningful interpretation. Henceforth, we reiterate the need for using our proposed robust radiomics analysis framework for addressing differences in IPR. Furthermore, the workflow can be generalized to evaluate other harmonization methods.

## 5. Conclusions

The reproducibility of a given HRF, and its harmonizabilty with ComBat are not constants, but they depend on the degree of variation in a single reconstruction parameter (the in-plane resolution) of the scans being analyzed. This implies that additional changes in the acquisition and reconstruction parameters could further reduce the number of reproducible and harmonizable HRFs. When scans acquired with different IPR values are to be analyzed, resampling the scans to a unified resolution can significantly improve the reproducibility of HRFs. Interpolation methods (CWS, HWS, BWS, WWS, and B-spline) were found to be superior to ComBat harmonization alone in addressing the variations in HRFs attributed to differences in IPR, and the combination of an IM with ComBat following NUIR could increase the number of reproducible HRFs in some scenarios. The application of our proposed framework aids the selection of data- and outcome-specific interpolation and harmonization methods, and it is expected to improve the translation and generalizability of radiomics analyses.

## Figures and Tables

**Figure 1 cancers-13-01848-f001:**
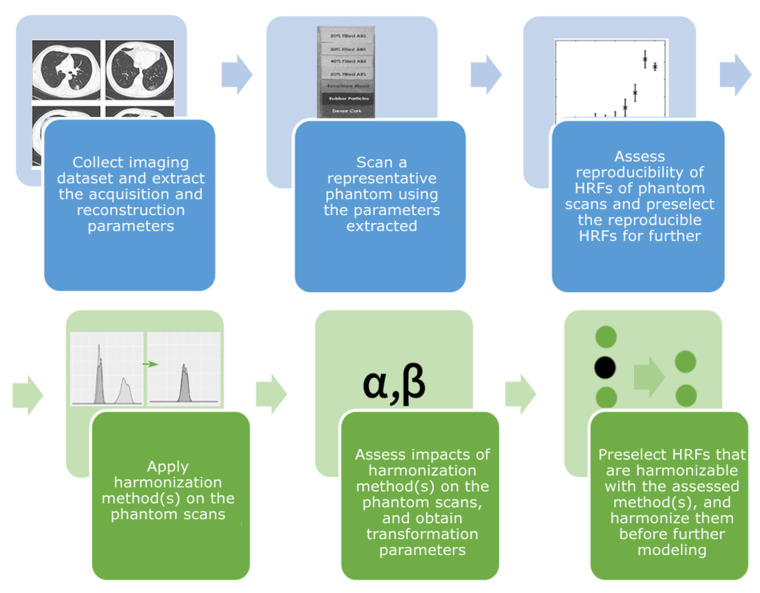
Proposed reproducible radiomic analysis workflow.

**Figure 2 cancers-13-01848-f002:**
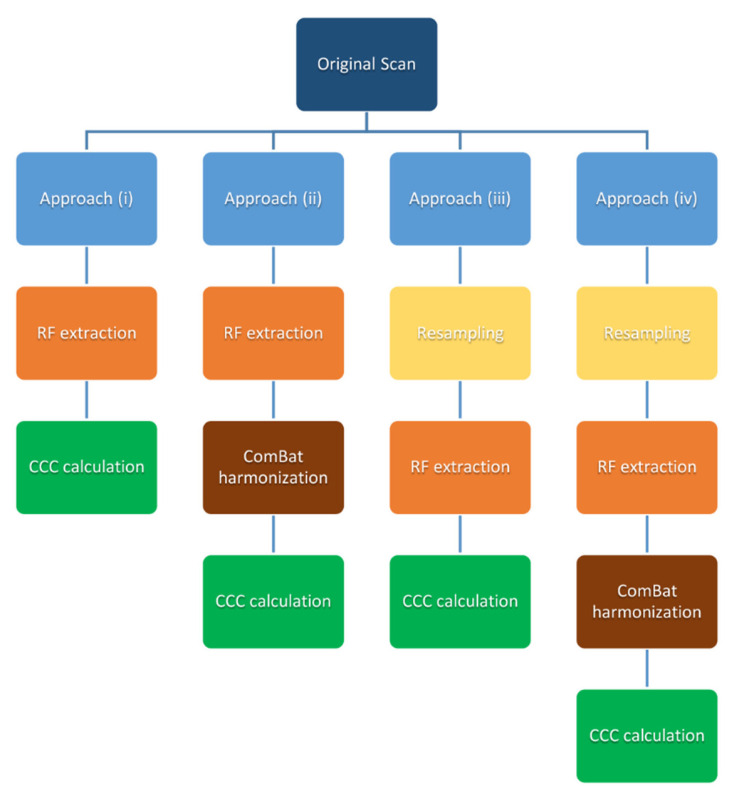
Reproducibility analysis approaches.

**Figure 3 cancers-13-01848-f003:**
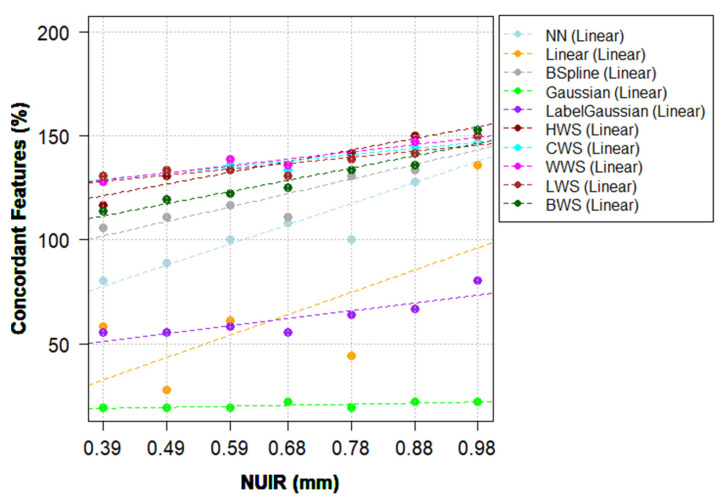
The percentage of concordant HRFs following resampling compared to no resampling with linear trendlines, Discovery STE model.

**Figure 4 cancers-13-01848-f004:**
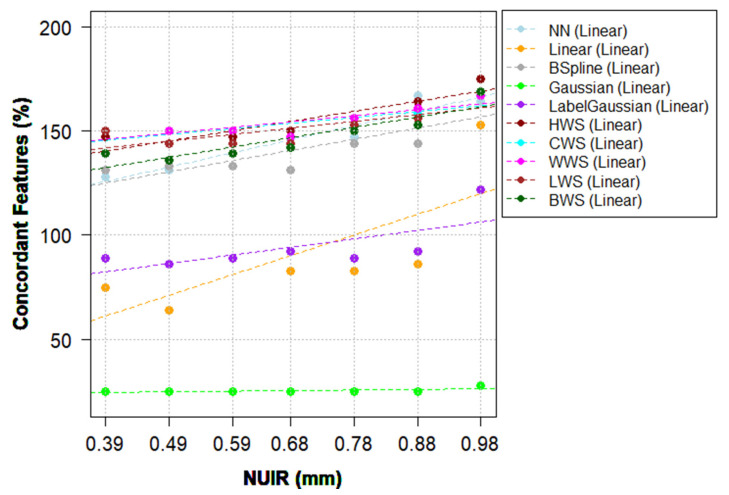
The percentage of concordant HRFs following resampling and ComBat harmonization compared to no resampling with linear trendlines, Discovery STE model.

**Table 1 cancers-13-01848-t001:** Scanning parameters of the phantom data.

Scanner		Pixel Spacing (mm^2^)
Discovery STE	LightSpeed Pro 32	
CCR-2-001	CCR-2-022	0.39 × 0.39
CCR-2-002	CCR-2-023	0.49 × 0.49
CCR-2-003	CCR-2-024	0.59 × 0.59
CCR-2-004	CCR-2-025	0.68 × 0.68
CCR-2-005	CCR-2-026	0.78 × 0.78
CCR-2-006	CCR-2-027	0.88 × 0.88
CCR-2-007	CCR-2-028	0.98 × 0.98

**Table 2 cancers-13-01848-t002:** Number of pairwise concordant handcrafted radiomic features (HRFs) with a concordance correlation coefficient (CCC) > 0.9 before resampling, Discovery STE model.

Scan	CCR-2-001	CCR-2-002	CCR-2-003	CCR-2-004	CCR-2-005	CCR-2-006
CCR-2-002	75 (82.4%)					
CCR-2-003	57 (62.6%)	78 (85.7%)				
CCR-2-004	53 (58.2%)	64 (70.3%)	83 (91.2%)			
CCR-2-005	50 (54.9%)	61 (67.0%)	72 (79.1%)	86 (94.5%)		
CCR-2-006	51 (56.0%)	58 (63.7%)	68 (74.7%)	76 (83.5%)	85 (93.4%)	
CCR-2-007	39 (42.9%)	42 (46.2%)	44 (48.4%)	52 (57.1%)	60 (64.9%)	83 (91.2%)

**Table 3 cancers-13-01848-t003:** Number of pairwise concordant HRFs with a CCC > 0.9 after ComBat harmonization, Discovery STE model.

Scan	CCR-2-001	CCR-2-002	CCR-2-003	CCR-2-004	CCR-2-005	CCR-2-006
CCR-2-002	79 (86.8%)					
CCR-2-003	65 (71.4%)	79 (86.8%)				
CCR-2-004	59 (64.8%)	70 (76.9%)	83 (91.2%)			
CCR-2-005	58 (63.7%)	66 (72.5%)	75 (82.4%)	87 (95.6%)		
CCR-2-006	57 (62.6%)	65 (71.4%)	70 (76.9%)	84 (92.3%)	86 (94.5%)	
CCR-2-007	48 (52.7%)	55 (60.4%)	57 (62.6%)	60 (65.9%)	73 (80.2%)	84 (92.3%)

**Table 4 cancers-13-01848-t004:** Number of pairwise concordant HRFs with a CCC > 0.9 after resampling * using CosineWindowedSinc (CWS), Discovery model.

Scan	CCR-2-001	CCR-2-002	CCR-2-003	CCR-2-004	CCR-2-005	CCR-2-006
CCR-2-002	89 (97.8%)					
CCR-2-003	86 (94.5%)	88 (96.7%)				
CCR-2-004	86 (94.5%)	85 (93.4%)	88 (96.7%)			
CCR-2-005	86 (94.5%)	88 (96.7%)	91 (100%)	89 (97.8%)		
CCR-2-006	78 (85.7%)	77 (84.6%)	83 (91.2%)	79 (86.8%)	88 (96.7%)	
CCR-2-007	53 (58.2%)	53 (58.2%)	55 (60.4%)	54 (59.3%)	60 (65.9%)	85 (93.4%)

* All scans were resampled to the median pixel spacing value (0.49 × 0.49 mm^2^).

**Table 5 cancers-13-01848-t005:** Number of pairwise concordant HRFs with a CCC > 0.9 after ComBat following resampling * using CWS, Discovery STE model.

Scan	CCR-2-022	CCR-2-023	CCR-2-024	CCR-2-025	CCR-2-026	CCR-2-027
CCR-2-023	89 (97.8%)					
CCR-2-024	86 (94.5%)	88 (96.7%)				
CCR-2-025	86 (94.5%)	85 (93.4%)	88 (96.7%)			
CCR-2-026	86 (94.5%)	88 (96.7%)	91 (100%)	89 (97.8%)		
CCR-2-027	79 (86.8%)	78 (85.7%)	84 (92.3%)	84 (92.3%)	89 (97.8%)	
CCR-2-028	57 (62.6%)	61 (67.0%)	60 (65.9%)	59 (64.8%)	72 (79.1%)	85 (93.4%)

* All scans were resampled to the median pixel spacing value (0.49 × 0.49 mm^2^).

## Data Availability

The data presented in this study are openly available on TCIA.org at http://doi.org/10.7937/K9/TCIA.2017.zuzrml5b (accessed on 7 January 2021).

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
