# Peer review of "The Effects of In-Plane Spatial Resolution on CT-Based Radiomic Features’ Stability with and without ComBat Harmonization"

_cancers, 2021, doi:10.3390/cancers13081848_

Round 1
Reviewer 1 Report
The dependence of radiomic features on imaging parameters is a fundamental problem underlying the definition of any predictive model. The proposed study is well placed, with a clear and exhaustive treatment, and the choices made are motivated/justified. Only a few comments to approach appropriately, in order to make the study even more robust.
Main comments
Comment 1. A work showing an interesting processing and normalization pipeline of radiomic features is used in [Papanikolaou N, Matos C, Koh DM (2020) How to develop a meaningful radiomic signature for clinical use in oncologic patients. Cancer Imaging 20:33]. It would certainly be interesting that the authors show how the framework can be integrated into clinical routine.
Comment 2. The authors based their work on the acquisitions obtained from a phantom, which is certainly useful if an approach has to be developed and calibrated, but it certainly represents a simplified scenario with respect to human images. It would be interesting to have the point of view of the authors and what they think they do about it (also as future work).
Comment 3. In clinical practice we do not necessarily go down to the level of resolution or interpolation methods. These parameters are intrinsic to the machine, and the radiologist (or technician) finds himself using a specific imaging protocol in relation to what he wants to highlight. It would be interesting to evaluate the stability of the features considering additional parameters of CT imaging, but probably with a dataset ready and available online like the one used, you don't have the possibility to do so. Is it possible to have the point of view of the authors?
Minor comments
- page 1, section 'Simple Summary': "HRFS"-> "HRFs".
- page 2, in order to enhance readability, the font size in figure 1 needs to be increased.
Author Response
Cancers-1179116: “The effects of in-plane spatial resolution on CT-based radiomic features’ stability with and without ComBat harmonization”
We would like to thank the editor and reviewers for their comments, which have ultimately led to an overall improvement of the manuscript. We have responded to each comment separately, and amended the manuscript accordingly. Please find our responses below. Line numbers refer to the revised manuscript.
Reviewer 1
The dependence of radiomic features on imaging parameters is a fundamental problem underlying the definition of any predictive model. The proposed study is well placed, with a clear and exhaustive treatment, and the choices made are motivated/justified. Only a few comments to approach appropriately, in order to make the study even more robust.
Comment 1: “A work showing an interesting processing and normalization pipeline of radiomic features is used in [Papanikolaou N, Matos C, Koh DM (2020) How to develop a meaningful radiomic signature for clinical use in oncologic patients. Cancer Imaging 20:33]. It would certainly be interesting that the authors show how the framework can be integrated into clinical routine.”
Our response: We thank the reviewer for the comment. We added the study to the list of referenced manuscripts in the introduction section. We further elaborated on how our proposed framework will help including radiomic signatures in clinical practice:
Pages 12-13, lines 485-488: “Our proposed framework addresses this issue, and guides the selection of reproducible and harmonizable HRFs before developing a radiomic signature, which helps the translation and generalization of results, and ultimately the inclusion of radiomic signatures in clinical practice.”
Comment 2: “The authors based their work on the acquisitions obtained from a phantom, which is certainly useful if an approach has to be developed and calibrated, but it certainly represents a simplified scenario with respect to human images. It would be interesting to have the point of view of the authors and what they think they do about it (also as future work).”
Our response: We thank the reviewer for the comment. We agree with the reviewer. We have added the following to the manuscript:
Page 13, lines 503-510: “Furthermore, while phantom studies reflect on the reproducibility of HRFs extracted from anthropomorphic phantoms, HRFs extracted from human tissue are expected to have a wider range of variations, due to the inclusion of biologic factors. This knowledge, combined with our findings, necessitate the critical investigation of the reproducibility of HRFs across the different scanning parameters/scanners before performing any statistical analysis, and future investigations into the effects of differences in acquisition and reconstruction parameters on the reproducibility of HRFs extracted from human tissues, if feasible.”
Comment 3: “In clinical practice we do not necessarily go down to the level of resolution or interpolation methods. These parameters are intrinsic to the machine, and the radiologist (or technician) finds himself using a specific imaging protocol in relation to what he wants to highlight. It would be interesting to evaluate the stability of the features considering additional parameters of CT imaging, but probably with a dataset ready and available online like the one used, you don't have the possibility to do so. Is it possible to have the point of view of the authors?”
Our response: We thank the reviewer for the comment. We agree that in practice, the majority of retrospective datasets have more variations in the acquisition and reconstruction parameters. However, we performed this study from a technical standpoint. We analysed this dataset to show that the range of variation in a single parameter has different effects on radiomic features in a way that we can interpret the findings of. It would certainly be of great interest to perform an overarching analysis on scans acquired with all the different combinations on all different imaging vendors available for clinical use, as it would allow the definition of reproducible and harmonizable features. However, due to the lack of such data, we proposed our framework to guide radiomic analysis until such a dataset could be collected. We reflected upon that in the manuscript:
Page 12, lines 492-500: “Some HRFs were not found to be concordant on one of the scanner models before or after resampling to any of the available NUIRs using any of the IMs, but were found to be concordant on the other scanner model. Other HRFs were found to be concordant across different scanner models and IPRs. These findings indicate the need for performing reproducibility studies depending on the data under study, and the fact that at this level, we are unable to provide a list of HRFs that can be used regardless of the acquisition and reconstruction parameters and scanner models used. However, it lays down the bases for identifying reproducible HRFs before performing data analysis.”
Page 14, lines 516-520: “The reproducibility of a given HRF, and its harmonizabilty with ComBat are not constants, but depended on the degree of variation in a single reconstruction parameter (the in-plane resolution) of the scans being analyzed. This implies that additional changes in the acquisition and reconstruction parameters could further reduce the number of reproducible and harmonizable HRFs.”
Comment 4: “page 1, section 'Simple Summary': "HRFS"-> "HRFs".”
Our response: We thank the reviewer for the comment. The type error has been corrected.
Comment 5: “page 2, in order to enhance readability, the font size in figure 1 needs to be increased.”
Our response: We thank the reviewer for the comment. The font size in figure 1 has been increased.

Reviewer 2 Report
This work analyzes the effect of in-plane spatial resolution (IPR) on the stability of radiomic features, which is evaluated by Lin's concordance correlation coefficient (CCC). Both resampling techniques and ComBat harmonization are tested. The analysis is done by using CCR phantom data from the well-known TCIA dataset. The work is definitely interesting and relevant in the field. I have the following comments:
- Section 2.1. Among scanner parameters, please also report tube voltage if available.
- A comment about the definition of ROI segmentation masks with different pixel spacing would be appreciated.
- Section 2.3. Page 5, line 185 reports that 70 ROIs are considered. Please clarify whether this number is obtained multiplying the number of independent ROIs (10, line 177) by the number of scans (7, with different IPRs), or it has a different meaning.
- As reported in page 11, line 389, feature names are not directly relevant for the purpose of this paper. However, they may be useful to reproduce results. The complete list of considered radiomic features should be inserted as supplementary material.
- Section 2.4. There are several formatting problems in equations (1) and (2). Moreover, symbols inside text have different fonts. Please fix/rewrite them in an accurate way, also using subscripts/superscripts.
- Section 2.5. Please add details about the set of paired readings used to compute CCC, i.e., the precise definition of vectors x,y to calculate CCC(x,y) for each radiomic feature. Moreover, all parameters of ComBat method should be reported (e.g., possible use of reference batch, parametric adjustment, etc.).
Author Response
Cancers-1179116: “The effects of in-plane spatial resolution on CT-based radiomic features’ stability with and without ComBat harmonization”
We would like to thank the editor and reviewers for their comments, which have ultimately led to an overall improvement of the manuscript. We have responded to each comment separately, and amended the manuscript accordingly. Please find our responses below. Line numbers refer to the revised manuscript.
Reviewer 2:
This work analyzes the effect of in-plane spatial resolution (IPR) on the stability of radiomic features, which is evaluated by Lin's concordance correlation coefficient (CCC). Both resampling techniques and ComBat harmonization are tested. The analysis is done by using CCR phantom data from the well-known TCIA dataset. The work is definitely interesting and relevant in the field. I have the following comments:
Comment 1: “Section 2.1. Among scanner parameters, please also report tube voltage if available.”
Our response: We thank the reviewer for the comment. We added the tube voltage to the description.
Page 4, line 146: “…,tube voltage (120 kV),..”
Comment 2: “A comment about the definition of ROI segmentation masks with different pixel spacing would be appreciated.”
Our response: We thank the reviewer for the comment. We added the following to the manuscript:
Page 5, line 179: “… that occupy the same physical area of the phantom on each scan.”
Comment 3: “Section 2.3. Page 5, line 185 reports that 70 ROIs are considered. Please clarify whether this number is obtained multiplying the number of independent ROIs (10, line 177) by the number of scans (7, with different IPRs), or it has a different meaning.”
Our response: We thank the reviewer for the comment. We have rewritten the sentence to make it clearer:
Page 5, line 186: “ For each set of scans (7 scans, with 10 ROIs per scan)…”.
Comment 4: “As reported in page 11, line 389, feature names are not directly relevant for the purpose of this paper. However, they may be useful to reproduce results. The complete list of considered radiomic features should be inserted as supplementary material.”
Our response: We thank the reviewer for the comment. The list of reproducible HRFs on both scanner models have been added as supplementary lists S1 and S2, as well as the lists of non-highly correlated reproducible HRFs (Lists S3 and S4).
Comment 5: “Section 2.4. There are several formatting problems in equations (1) and (2). Moreover, symbols inside text have different fonts. Please fix/rewrite them in an accurate way, also using subscripts/superscripts.”
Our response: We thank the reviewer for the comment. The equations have been rewritten.
Comment 6: “Section 2.5. Please add details about the set of paired readings used to compute CCC, i.e., the precise definition of vectors x,y to calculate CCC(x,y) for each radiomic feature. Moreover, all parameters of ComBat method should be reported (e.g., possible use of reference batch, parametric adjustment, etc.).”
Our response: We thank the reviewer for the comment. We have added further clarification for the CCC calculation and ComBat settings.
Page 6, line 231-236: “In each run, the CCC was calculated between a different pair of scans. For (ii), HRFs with nearly zero variance (i.e HRFs which have the same value in 95% or more of the data points) had to be removed before applying ComBat. Parametric prior estimations were used, and no reference batch was assigned for ComBat application. The CCC was calculated after harmonizing the remaining HRFs using ComBat. In each run, ComBat was applied in two batches (scans).”

Reviewer 3 Report
The manuscript proposes an in-depth evaluation of the effects of the in-plane resolution, the different interpolation methods and the ComBat harmonization on radiomic features reproducibility, using phantom CT images. The evaluation was carried out in a valid way, by considering the combination of different factors together and a general framework for features reproducibility evaluation was finally proposed. The obtained results were generally in agreement with the published literature.
Although the subject and the results are not completely novel and the literature in this field is becoming increasingly wider, I think that the work is valuable and it contains some novelties, thanks to the large range of different interpolation methods evaluated and the to the analysis of the combined effects of the different factors. In general, the manuscript is well written, and I particularly appreciated the results schematization as an effort of simplifying their presentation.
I have some comments to the authors about the analysis:
- It is now generally acknowledged that some radiomic features are strongly affected by ROI volume, and that there are some indices that particularly embed this information. It was recently reported that some radiomic models, also already validated, should be revised as they included only volumetric information, hidden by other textural features that would not contain additional information [Welch et al, Radiotherapy and Oncology, 2019. doi: 10.1016/J.RADONC.2018.10.027]. I’m wondering if the authors have considered this aspect in their analysis. If I correctly understood, the analyzed ROI had the same volume. However, as it is reported in the cited work, there are features that remained reproducible even if the intensity matrix changed, just because they were merely a surrogate of volume, that was obviously invariant independently on pixel size and interpolation methods. I suggest a preliminary analysis on the relation between features and volume, in order to discard the most correlated ones.
- Another point that I think should be discussed is the strong correlation that may exist between features themselves, especially between those extracted from the same texture matrix. Even in this case, maybe an additional analysis to correct for the number of correlated features could improve the results.
Minor comments are:
- I suggest to include in the discussion other recent works treating the same topic, e.g. Ligero et al, 2021, European Radiology, doi: https://doi.org/10.1007/s00330-020-07174-0; Shin-Hyung et al, 2021, Cancer Imaging, doi: https://doi.org/10.1186/s40644-021-00388-5.
- Finally, I suggest adding some more details about the analyzed phantom images. Maybe a figure could be of help.
Author Response
Cancers-1179116: “The effects of in-plane spatial resolution on CT-based radiomic features’ stability with and without ComBat harmonization”
We would like to thank the editor and reviewers for their comments, which have ultimately led to an overall improvement of the manuscript. We have responded to each comment separately, and amended the manuscript accordingly. Please find our responses below. Line numbers refer to the revised manuscript.
Reviewer 3
The manuscript proposes an in-depth evaluation of the effects of the in-plane resolution, the different interpolation methods and the ComBat harmonization on radiomic features reproducibility, using phantom CT images. The evaluation was carried out in a valid way, by considering the combination of different factors together and a general framework for features reproducibility evaluation was finally proposed. The obtained results were generally in agreement with the published literature.
Although the subject and the results are not completely novel and the literature in this field is becoming increasingly wider, I think that the work is valuable and it contains some novelties, thanks to the large range of different interpolation methods evaluated and the to the analysis of the combined effects of the different factors. In general, the manuscript is well written, and I particularly appreciated the results schematization as an effort of simplifying their presentation.
I have some comments to the authors about the analysis:
Comment 1: It is now generally acknowledged that some radiomic features are strongly affected by ROI volume, and that there are some indices that particularly embed this information. It was recently reported that some radiomic models, also already validated, should be revised as they included only volumetric information, hidden by other textural features that would not contain additional information [Welch et al, Radiotherapy and Oncology, 2019. doi: 10.1016/J.RADONC.2018.10.027]. I’m wondering if the authors have considered this aspect in their analysis. If I correctly understood, the analyzed ROI had the same volume. However, as it is reported in the cited work, there are features that remained reproducible even if the intensity matrix changed, just because they were merely a surrogate of volume, that was obviously invariant independently on pixel size and interpolation methods. I suggest a preliminary analysis on the relation between features and volume, in order to discard the most correlated ones. Another point that I think should be discussed is the strong correlation that may exist between features themselves, especially between those extracted from the same texture matrix. Even in this case, maybe an additional analysis to correct for the number of correlated features could improve the results.
Our response: We thank the reviewer for the comments. We agree with the reviewer that the correlation with volume and the HRFs among themselves is an important issue to address. We now added the assessment of correlation between all the HRFs, including volume to the methods section, and reported the number of highly correlated HRFs in between those identified as reproducible and harmonizable with ComBat in the results section, with a reflection on the findings in the discussion. We also added the lists of non-highly correlated HRFs on both scanner models to the supplementary materials (Lists S3 and S4). As these changes were extensive, we have not included them in this response. Please see the highlighted changes in the document.
Comment 2: “I suggest to include in the discussion other recent works treating the same topic, e.g. Ligero et al, 2021, European Radiology, doi: https://doi.org/10.1007/s00330-020-07174-0; Shin-Hyung et al, 2021, Cancer Imaging, doi: https://doi.org/10.1186/s40644-021-00388-5.”
Our response: We thank the reviewer for the comment. We included some of the mentioned studies in the discussion.
Page 12, lines 437-448: “A previous study investigated the effects on HRFs of voxel size resampling using linear interpolation. The authors resampled the scans of a phantom to a single voxel size, which was larger than the largest voxel size in the original scans, and reported that around 20% of the HRFs (N=213) became concordant after resampling [22]. Another study also investigated the effects of voxel size on HRFs of lung cancer patients [20]. The authors resampled all the scans to a single common voxel size using linear interpolation, and reported that resampling does not eliminate all the variations in feature values even when the only variation in scan acquisition and reconstruction parameters was the voxel size, but is favorable to no resampling. Another group investigated the effects of variation in several acquisition and reconstruction parameters on a 13 layer phantom using a different approach, and reported that resampling the scans to isotropic voxels increased the percentage of concordant HRFs from 59.5% to 89.3% [41].”
Comment 3: Finally, I suggest adding some more details about the analyzed phantom images. Maybe a figure could be of help.
Our response: We thank the reviewer for the comment. The phantom, which is widely available, consists of 10 cartridges: “The first four cartridges are 3D printed ABS plastic with 20%, 30%, 40%, and 50% honeycomb fill, and they provide regular, periodic textures. The next three cartridges provide natural textures: sycamore wood, cork, and extra dense cork. A cartridges of shredded rubber particles provides textures similar to those of non-small cell lung cancer. The ninth cartridge is solid, homogenous acrylic and provides a minimal texture control. Finally, the 10th cartridge is 3D printed plaster has the highest electron density (400 – 600 HU) and is intended to more similar to bone” (TCIA.org). We added a figure describing the phantom in the supplementary materials (Figure S1).
